# Neuroprotective Effects of Tripeptides—Epigenetic Regulators in Mouse Model of Alzheimer’s Disease

**DOI:** 10.3390/ph14060515

**Published:** 2021-05-27

**Authors:** Vladimir Khavinson, Anastasiia Ilina, Nina Kraskovskaya, Natalia Linkova, Nina Kolchina, Ekaterina Mironova, Alexander Erofeev, Michael Petukhov

**Affiliations:** 1Saint Petersburg Institute of Bioregulation and Gerontology, 197110 Saint Petersburg, Russia; vladimir@khavinson.ru (V.K.); miayy@yandex.ru (N.L.); katrine1994@mail.ru (E.M.); 2Pavlov Institute of Physiology of Russian Academy of Sciences, 199034 Saint Petersburg, Russia; 3Institute of Biomedical Systems and Biotechnology, Peter the Great St. Petersburg State Polytechnic University, 195251 Saint Petersburg, Russia; ninakraskovskaya@gmail.com (N.K.); alexandr.erofeew@gmail.com (A.E.); 4Petersburg Nuclear Physics Institute Named after B.P. Konstantinov, NRC “Kurchatov Institute”, 188300 Gatchina, Russia; ininakolchina@mail.ru (N.K.); michael.petukhov@yandex.ru (M.P.); 5Russian Scientific Center of Radiology and Surgical Technologies Named after A.M. Granov, 197758 Saint Petersburg, Russia

**Keywords:** EDR peptide, KED peptide, epigenetic regulation, Alzheimer’s disease, gender, neuronal dendritic spines, neuroplasticity, 5xFAD mice

## Abstract

KED and EDR peptides prevent dendritic spines loss in amyloid synaptotoxicity in in vitro model of Alzheimer’s disease (AD). The objective of this paper was to study epigenetic mechanisms of EDR and KED peptides’ neuroprotective effects on neuroplasticity and dendritic spine morphology in an AD mouse model. Daily intraperitoneal administration of the KED peptide in 5xFAD mice from 2 to 4 months of age at a concentration of 400 μg/kg tended to increase neuroplasticity. KED and EDR peptides prevented dendritic spine loss in 5xFAD-M mice. Their action’s possible molecular mechanisms were investigated by molecular modeling and docking of peptides in dsDNA, containing all possible combinations of hexanucleotide sequences. Similar DNA sequences were found in the lowest-energy complexes of the studied peptides with DNA in the classical B-form. EDR peptide has binding sites in the promoter region of *CASP3*, *NES*, *GAP43*, *APOE*, *SOD2*, *PPARA*, *PPARG*, *GDX1* genes. Protein products of these genes are involved in AD pathogenesis. The neuroprotective effect of EDR and KED peptides in AD can be defined by their ability to prevent dendritic spine elimination and neuroplasticity impairments at the molecular epigenetic level.

## 1. Introduction

Alzheimer’s disease (AD) is a common neurodegenerative disease in the elderly and the most prevalent cause of dementia. AD is characterized by a progressive cognitive impairment [1]. The prevalence of AD globally is increasing exponentially [2], and the number of people with dementia is predicted to increase to 131.5 million worldwide by 2050 [3]. The most common form of AD is the late-onset AD (LOAD), defined as the AD with an onset at the age over 65, while the early-onset AD (EOAD) accounts for approximately 1% to 6% of all cases. The onset age of EOAD ranges roughly from 30 to 65 years. Genetically the disease is divided into sporadic cases (SAD) and familial cases (FAD). SAD results from a combination of genetic and environmental factors [4]. FAD is associated with mutations in one of the three genes: amyloid precursor protein (APP), presenilin 1 (PSEN1), presenilin 2 (PSEN2).

The APP gene comprises 18 exons, and alternative splicing can give rise to 10 different isoforms consisting of 563 to 770 amino acid residues. The processing of APP by β-secretase activity followed by gamma-secretase cleavage produces different Aβ-fragments, called Aβ 1-40 and Aβ 1-42. These fragments are neurotoxic forms due to their propensity to rapidly aggregate in amyloid plaques—one of the histopathological signs [4].

Although amyloid plaques are the main histopathological signs of AD, the level of amyloid deposits is poorly correlated with cognitive impairments [5]. At the same time, the synaptic loss is one of the earliest signs of AD, closely related to cognitive impairment [6]. Postsynaptic structures of dendrites are called dendritic spines, divided into subclasses such as stubby, thin and mushroom by morphological features [7]. Stubby spines are characterized by the absence of the neck and are dominant at the early stages of postnatal development. Stubby spines are poorly presented in adulthood since they result from the elimination of mushroom spines [7]. Thin spines with a small head and a long thin neck are dynamic postsynaptic structures, considered as “spines of learning” that are involved in forming new memories [8]. Mushroom spines are stable postsynaptic structures with a large head and a fine neck [9]. Mushroom spines form the most active synapses, contain a larger amount of postsynaptic density (PSD) and receptors, and are considered “memory spines”. Hippocampal mushroom spines are strongly eliminated in AD. The loss of mushroom spines may underlie cognitive impairments during AD progression [10]. In neurodegenerative diseases such as AD, the thin spines’ number increases proportionally to the mushroom spines’ elimination [11,12,13]. LTP (long-term potentiation) is the physiological basis of neuroplasticity, which underlies learning and memory [14,15]. LTP represents increased synaptic transmission between the two neurons, which persists for a long time when the synaptic pathway is affected. LTP is mediated by synthesizing new protein molecules in the postsynaptic terminal, causing morphological changes in dendritic spines, thus reflecting functional changes in synapses [6].

Despite extensive long-term studies, pharmacological correction of main histopathological signs of AD, such as β-amyloid deposition, produced no productive results [1]. Since the precise molecular mechanisms of AD pathogenesis remain elusive, there is an urgent need for supportive agents—safe and non-addictive neuroprotective remedies that are effective at the early stages of cognitive decline. In the context of such a strategy, the short peptide may have therapeutic value in AD treatment.

The first described neuroprotective peptide drugs derived from the cerebral cortex of cattle and pigs were Cortexin [16] and Cerebrolysine [17]. According to a multicenter randomized placebo-controlled research, Cortexin effectively treated chronic cerebral ischemia [18]. Cortexin contributed to memory restoration in patients with acute ischemic stroke [19]. The Shanghai Jiao Tong University (China) conducted a meta-analysis of the Cerebrolysin effectiveness in the AD treatment. Cerebrolysin administration resulted in clinically relevant improvements in cognitive function, non-cognitive psychiatric symptoms and daily activity in patients with mild and moderately severe stages of AD [17]. In a study of transgenic mThy1-hAPP751 mice AD model, intraperitoneal administration of Cerebrolysine at a dose of 5 mL/kg per day for 6 months reduced amyloid levels in the brain and reduced the synaptic pathology [20]. It is interesting that Semax is a fragment of adrenocorticotropin and was the first short peptide drug with neuroprotective properties described [21]. Semax has nootropic, psycho-stimulating, antioxidant and antihypoxic effects [22,23,24,25].

EDR peptide, Pinealon [26,27,28], is found in Cortexin and exhibits neuroprotective activity similar to this drug. Oral administration of EDR peptide and standard therapy improved memory in 59.4% of patients with craniocerebral trauma [29]. EDR peptide contributed to the psychoemotional state correction in the elderly by increasing the recovery index in the main group compared to the control [23]. EDR peptide restored the number of dendritic spines in medium spiny neurons in an in vitro model of Huntington’s disease (HD) [29]. EDR peptide also increased the number of mushroom spines up to 71% in primary hippocampal cultures treated with Aβ42 (amyloid synaptotoxicity AD model) [30]. In vitro and in vivo studies of the neuroprotective properties of the EDR peptide suggest that it can regulate gene expression and synthesis of proteins involved in AD pathogenesis [31].

KED peptide, Vesugen, is a vasoprotective peptide found in a polypeptide complex obtained from the cattle vessels [32,33,34]. Oral administration of the KED and EDR peptides contributed to improved cognitive functions in workers in hazardous working conditions [34]. KED peptide stimulated neuronal differentiation in human dental stem cells [35]. KED peptide increased the spine density up to 32% in an in vitro model of HD [29]. In amyloid synaptotoxicity, the KED peptide increased the mushroom spines number by 1.2 times [30].

This work aimed to study epigenetic mechanisms of EDR and KED peptides’ neuroprotective effects on neuroplasticity and dendritic spine morphology in a mouse model of AD. In the present study, we continued to address neuroprotective properties of EDR and KED short peptides in vivo by evaluating their effect by the calculation of the number and morphology of dendritic spines and neuroplasticity in the hippocampus of 5xFAD mice (B6SJL-Tg(APPSwFlLon, PSEN1*M146L*L286V)6799Vas/Mmjax, Stock No: 34840-JAX, PubMed: 17021169). 5xFAD mice had two transgenes in the genome: Swedish K670N/M671L, Florida I716V, London V717I mutations in the human APP gene and M146L and L286V mutations in the human PSEN1 gene. These mutations led to Aβ42 accumulation in amyloid plaques, synaptic loss and cognitive impairments [36].

## 2. Results

### 2.1. Effects of Tripeptides on LTP in Hippocampus of 5xFAD Mice in AD

LTP impairment in the hippocampus of 5xFAD mice at four months of age as previously described [37,38]. Based on literature data, LTP experiments were done in four-month-old 5xFAD mice and WT mice. In 5xFAD mice in hippocampal sections after high-frequency stimulation of Schaffer’s collaterals in the stratum radiatum region at the CA1-CA2 border, a tendency to impair neuroplasticity was revealed than wild-type mice. However, it has not reached the level of statistically significant differences (*p* = 0.057) between plasticity values in WT and 5xFAD mice (Figure 1). There was also a positive trend for the EDR and KED peptides to restore LTP in 5xFAD mice, although it has not reached the statistical significance level. (Figure 1a,b).

### 2.2. Effects of Tripeptides on the Neuron Morphology of 5xFAD-M Mice in AD

To access the dendritic spine morphology in the CA1 region of the hippocampus, we crossed 5xFAD mice with M mice to obtain 5xFAD-M mice. We founded the age-related spine density decreasing and mushroom spine loss in five-month-old 5xFAD (Figure 2, Figure 3 and Figure 4). It was consistent with the literature data on AD progression in 5xFAD mice [6,36,37]. Therefore, we concluded that 5xFAD-M mice could be a helpful AD model for evaluating the effects of peptides on neuron morphology.

The EDR peptide increased dendritic spine density of CA1 neurons in 5 month-old 5xFAD-M mice by 11% (*p* = 0.039) compared to the 5xFAD-M mice, injected with a physiological solution (SD was 11.31 ± 0.36 spines/10 μm and 12.64 ± 0.31 spines/10 μm in 5xFAD-M mice, injected with a physiological solution and EDR peptide, respectively) (Figure 2 and Figure 3). The EDR peptide restored spine density in 5 month-old 5xFAD-M mice up to the control level in M-line mice (*p* = 0.509) (SD was 12.89 ± 0.32 spines/10 μm in M mice, injected with a physiological solution). The EDR peptide did not affect the mushroom spine number (*p* = 0.053) that was increased by 12% (*p* = 0.00002) in 5 month-old 5xFAD mice compared to the control male M mice (MS was 44.57 ± 1.11%, 35.61 ± 1.64% and 39.15 ± 1.08% in M and 5xFAD-M mice, injected with a physiological solution and EDR peptide, respectively). The EDR peptide reduced the thin spine number by 10% (*p* = 0.024) (TS was 44.89 ± 1.65% and 50.15 ± 1.81% in 5xFAD-M mice, injected with a physiological solution and EDR peptide, respectively). The thin spine number increased by 19% (*p* = 0.00003) in 5xFAD mice, injected with a physiological solution, compared to the M mice (TS was 40.61 ± 1.26% in M mice, injected with a physiological solution) (Figure 2 and Figure 4). The EDR peptide did not affect the stubby spine number (*p* = 0.162), which did not differ in M-line and 5xFAD-M mice (*p* = 0.887) (SS was 14.62 ± 0.73%, 13.99 ± 0.84% and 15.90 ± 0.77% in M and 5xFAD-M mice, injected with a physiological solution and EDR peptide, respectively).

The neuroprotective effect of the KED peptide is different from that of the EDR peptide. The KED peptide did not alter dendritic spine density in 5 month-old 5xFAD-M mice (*p* = 0.103) (SD was 11.31 ± 0.36 spines/10 μm and 12.48 ± 0.28 spines/10 μm in 5xFAD-M mice, injected with a physiological solution and KED peptide, respectively) (Figure 2 and Figure 3). However, the KED peptide restored the number of mushroom spines in 5xFAD-M mice (*p* = 0.003) up to the control level (*p* = 0.157) in M mice (MS was 39.15 ± 1.08% and 41.25 ± 1.09% in 5xFAD-M mice, injected with a physiological solution and KED peptide, respectively) (Figure 2 and Figure 4). The KED-associated increase in mushroom spines number led to decreased thin spine number in 5xFAD-M mice by 13% (*p* = 0.008) up to the control level (*p* = 0.156) (TS was 43.84 ± 1.83% in 5xFAD-M mice, injected with the KED peptide) that was consistent with literature data [11] on the spine balancing in AD. The KED peptide did not affect the stubby spine number (*p* = 0.162) (SS was 14.97 ± 0.85% in 5xFAD-M mice injected with the KED peptide).

We conclude that systematic administration of the KED and EDR peptides prevent the elimination of postsynaptic structures in CA1 neuron of 5xFAD-M mice. The most pronounced effect is observed for the KED peptide due to the restoration of the thin and “memory” mushroom spines number up to the level of M mice, while the EDR peptide restores only the dendritic spine density. An interesting result is the absence of EDR and KED peptides’ effect on the stubby spines number amidst undetected differences in 5xFAD-M and M mice. It indicates the modulating effect of these peptides.

### 2.3. Sex-Related Differences in the Neuroprotective Effect of Short Peptides in 5xFAD-M Mice

5-months-old 5xFAD-M male and female mice showed differences in the total density and the mushroom spines number, while 5xFAD-M males had more pronounced spine pathology than females. Therefore, we investigated sex-related differences in the neuroprotective effect of short peptides in 5xFAD-M mice.

#### 2.3.1. Males

The EDR peptide increased dendritic spine density of CA1 neurons in 5 month-old male 5xFAD-M mice by 13% (*p* = 0.019) compared to the 5xFAD-M mice injected with a physiological solution (SD was 10.62 ± 0.49 spines/10 μm and 12.21 ± 0.40 spines/10 μm in 5xFAD-M mice, injected with a physiological solution and EDR peptide, respectively) (Figure 5 and Figure 6). Dendritic spine density of neurons in male 5xFAD-M mice, injected with the EDR peptide, was restored due to increased mushroom spines number by 25% (*p* = 0.004) compared to the control M mice (MS was 30.22 ± 2.78% and 40.23 ± 1.52% in 5xFAD-M mice, injected with a physiological solution and EDR peptide, respectively) The EDR peptide restored spine density and mushroom spines number (*p* = 0.883 and *p* = 0.197, respectively) in 5 month-old male 5xFAD-M mice up to the level of control M mice (SD and MS were 12.52 ± 0.38 spines/10 μm and 45.06 ± 1.64%, respectively, in M mice, injected with a physiological solution). The EDR peptide did not affect the thin spine number (*p* = 0.123) that was increased by 19% (*p* = 0.013) in male 5xFAD mice compared to the control male M mice (TS was 39.07 ± 1.78%, 48.21 ± 4.20% and 42.01 ± 1.96% in male M and 5xFAD-M mice injected with a physiological solution and EDR peptide, respectively). The EDR peptide did not affect the stubby spine number (*p* = 0.998), which did not change in male M and 5xFAD-M mice (*p* = 0.972) (SS was 15.61 ± 1.02%, 16.15 ± 1.45% and 15.93 ± 1.07% in male M and 5xFAD-M mice, injected with a physiological solution and EDR peptide, respectively) (Figure 5 and Figure 7).

The KED peptide increased dendritic spine density of CA1 neurons in 5 month-old male 5xFAD-M mice by 22% (*p* = 0.00001) compared to the 5xFAD-M mice, injected with a physiological solution (SD was 10.62 ± 0.49 spines/10 μm and 13.56 ± 0.27 spines/10 μm in 5xFAD-M mice, injected with a physiological solution and KED peptide, respectively) (Figure 5 and Figure 6). Dendritic spine density of neurons in males of 5xFAD-M mice, injected with the KED peptide, was restored due to the increase in mushroom spines number by the 27% (*p* = 0.030) compared to the control M mice (MS was 30.22 ± 2.77% and 41.13 ± 1.48% in 5xFAD-M mice, injected with a physiological solution and KED peptide, respectively) The KED peptide restored spine density and mushroom spines number (*p* = 0.131 and *p* = 0.283, respectively) in 5 month-old male 5xFAD-M mice up to the control level. The KED peptide did not affect the thin (*p* = 0.348) and stubby (*p* = 0.448) spine number (TS and SS were 43.61 ± 1.46% and 14.16 ± 0.85%, respectively, in male 5xFAD-M mice, injected with the KED peptide) (Figure 5 and Figure 7).

Thus, there was a detectable reduction in the dendritic spines’ density of the CA1 neurons in the five-month-old 5xFAD-M males due to the elimination of mushroom spines involved in memory storage. Systematic administration of the EDR and KED peptides restored the mushroom (but not thin) spines number up to the control level, increasing the total spine density in male 5xFAD-M mice.

#### 2.3.2. Females

The EDR peptide increased dendritic spine density of CA1 neurons in 5 month-old female 5xFAD-M mice by 12% (*p* = 0.035) compared to the 5xFAD-M mice, injected with a physiological solution (SD was 11.74 ± 0.49 spines/10 μm and 13.31 ± 0.51 spines/10 μm in 5xFAD-M mice, injected with a physiological solution and EDR peptide, respectively) (Figure 8 and Figure 9). The EDR peptide restored spine density (*p* = 0.978) in 5 month-old female 5xFAD-M mice up to the control level (SD was 13.39 ± 0.53 spines/10 μm in M mice were injected with a physiological solution). However, the EDR peptide did not affect (*p* = 0.681) on mushroom spine number that was decreased by 19% (*p* = 0.0004) in 5 month-old female 5xFAD-M mice compared to M mice (MS was 43.94 ± 1.44%, 35.60 ± 1.62% and 36.35 ± 1.43% in female M and 5xFAD-M mice, injected with a physiological solution and EDR peptide, respectively). The EDR peptide did not affect the thin spine number (*p* = 0.517) that was increased by 17% (*p* = 0.004) in female 5xFAD-M mice compared to the control female M mice (TS was 42.65 ± 1.69%, 51.38 ± 1.76% and 48.45 ± 1.86% in female M and 5xFAD-M mice, injected with a physiological solution and EDR peptide, respectively). The EDR peptide did not affect the stubby spine number (*p* = 0.061) that was similar in female M and 5xFAD-M mice (*p* = 0.935) (SS was 13.31 ± 0.98%, 12.62 ± 1.05% and 15.86 ± 1.01% in female M and 5xFAD-M mice, injected with a physiological solution and EDR peptide, respectively) (Figure 8 and Figure 10).

The KED peptide decreased the thin spine number by 14% (*p* = 0.028) compared to the 5xFAD-M mice, injected with a physiological solution (TS was 44.18 ± 2.99% in 5xFAD-M mice, injected with the KED peptide). The KED peptide restored the thin spine number in 5 month-old female 5xFAD-M mice up to the control level in M mice (*p* = 0.618). The KED peptide did not affect dendritic spine density (11.24 ± 0.33 spines/10 μm, *p* = 0.973), mushroom (39.66 ± 2.09%, *p* = 0.109) and stubby (16.12 ± 1.50%, *p* = 0.062) spines number of CA1 neurons in 5 month-old female 5xFAD-M mice.

Thus, the EDR peptide restored dendritic spine density in female 5xFAD-M mice. The KED peptide restored the thin (but not mushroom) spines number in female 5xFAD-M mice up to the control level.

### 2.4. Possible Molecular Mechanism of KED and EDR Neuroprotective Activity

To explain the possible mechanism of action of the tripeptides, with the help of the methods of molecular docking of ligands, we studied their binding to all possible dsDNA sequences of 6 nucleotides in length, generating unique spatial structures of the double helix in the classical B-form. In total, there are 2080 such DNA sequences, which are listed in Table 1, Figure 11, panels a and b show the spatial structures of the most low-energy complexes of dsDNA with tripeptides EDR and KED (ICM score −44.7 and −26.6, respectively). As in many other cases [39], the positively charged residues of Lys^+^ and especially Arg^+^ at the C-terminus improve the binding of short peptides to dsDNA. Interestingly, both peptides under study bind in the same region of the minor groove of dsDNA, however, with an opposite arrangement of the N- and C-ends and rather similar, although not identical, dsDNA sequences.

One can see from the data that in the lowest energy complexes, EDR and KED peptides are rotated by 180 degrees relative to each other. This is obvious because the positively charged amino acids (Arg^+^ and Lys^+^, respectively) are placed at the C- and N-termini of the tripeptides. Despite such a huge difference in the positions of these peptides in the minor groove of DNA, the arrangement of charged, polar, and hydrophobic groups in these complexes is remarkably similar. Interestingly, the DNA sequences in these complexes are also very similar. Moreover, if we consider not the nucleotide sequences of the forward and reverse DNA strands but pairs of complementary bases GC and AT, the DNA structure in these complexes will be 100% identical to the EDR and KED peptides. It is noteworthy that the probability of such a random coincidence of the DNA structure in the best complexes for the studied peptides is (1/2)^6^ = 0.0156. Thus, such a coincidence is highly unlikely.

Such an observation—though still a very limited one—suggests that short peptides (and proteins) in complexes with dsDNA may recognize the sequences of base pairs instead of sequences of one or another DNA strand, as widely accepted. This has a good physical meaning since the bases are located near each other, and for favorable contacts between flexible peptides and DNA, it is not so important on which DNA strand the corresponding group is located if it is nearby.

This idea does not contradict the available experimental data on complexes of proteins with dsDNA and can even explain the specific binding of some proteins in the minor groove of dsDNA where, as is known, there is no possibility of unambiguous recognition of dsDNA sequences [40]. However, this approach may have many important consequences for interpreting the molecular mechanisms of interaction between proteins and DNA and, therefore, deserves a separate, more detailed consideration.

Table 1 presents statistical data on the binding of the studied EDR and KED peptides with all possible hex-nucleotide sequences of dsDNA in the classical B-form distributed over different clusters having the same mask, i.e., a sequence of base pairs AT&TA and GC&CG denoted W and S, respectively. The presented data demonstrate that for any DNA sequences, the EDR peptide has a much lower ICM score compared to the KED peptide, indicating much higher binding constants for complexes of the EDR peptide with dsDNA. In general, nucleotide sequences with a higher GC content have a higher affinity for the EDR and KED peptides, although there is no significant correlation between the high-affinity DNA sequences of these peptides. Since there are many high-affinity DNA sequences for the EDR and KED peptides with similar levels of their ICM scores, the binding of these peptides to dsDNA is likely to be of low selectivity. Interestingly, the best complexes for these two peptides have identical masks and highly homologous DNA sequences. Since the intracellular concentrations of these peptides are not known, this may explain similar biological functions of these peptides.

**Table 1 pharmaceuticals-14-00515-t001:** Distribution of ICM scores in the peptide—hexanucleotide complexes having the same DNA mask * sequence.

EDR Peptide				KED Peptide			
DNA Mask * Sequence	Mean ICM Score (STD)	Best ICM Score	Best DNA Sequence	DNA Mask * Sequence	Mean ICM Score (STD)	Best ICM Score	Best DNA Sequence
**SWSWSS**	−38.67 (3.24)	−44.71	**GAGTGG**	**SWSWSS**	−16.15 (3.57)	−26.56	**CAGAGG**
**WSSWSS**	−38.66 (2.35)	−44.01	**ACGTCG**	**WSWSWS**	−15.96 (2.94)	−24.02	**ACAGTG**
**SWSSWS**	−38.20 (3.50)	−42.76	**CACGTG**	**WSWSSW**	−15.91 (2.86)	−20.52	**TCTGGA**
**SSSWSS**	−38.02 (2.27)	−43.31	**GGGACG**	**WWSWSW**	−15.72 (2.68)	−22.32	**AACACT**
**WSSWSW**	−37.96 (2.25)	−43.78	**AGGAGT**	**SSWSWS**	−15.55 (3.91)	−22.54	**GCAGTG**
**SWSWWS**	−37.89 (2.66)	−44.43	**GAGTAC**	**WSWSWW**	−15.53 (3.12)	−22.33	**ACAGAA**
**SWSSSS**	−37.73 (2.61)	−42.76	**GTCCCC**	**SSSWSS**	−15.36 (2.93)	−20.72	**CGCTGG**
**WWSWSS**	−37.65 (2.87)	−43.04	**TACTCG**	**SSWSSS**	−15.34 (3.09)	−22.25	**CGTGGG**
**WSSWWS**	−37.45 (2.38)	−42.99	**TGGTTG**	**WSWSSS**	−15.31 (3.73)	−21.34	**AGTCCC**
**WWSSWS**	−37.36 (2.88)	−44.04	**AAGCTC**	**WWWSWS**	−15.28 (3.36)	−23.85	**TTTCAG**
**WSSSSS**	−37.34 (2.64)	−43.10	**AGGCCC**	**SWWSWS**	−15.27 (3.07)	−21.63	**GTTGTG**
**SSSSSS**	−37.31 (3.02)	−42.81	**CGGGCG**	**WWSWSS**	−15.26 (3.52)	−24.69	**TACAGG**
**SSSSWS**	−37.21 (2.37)	−41.61	**GGGGTG**	**SWWWSS**	−15.25 (2.84)	−23.29	**CAAACG**
**WSWSWW**	−37.17 (2.87)	−42.70	**AGAGTA**	**WSSWSS**	−15.23 (3.53)	−25.71	**AGCTCG**
**SSSWWS**	−37.14 (2.57)	−41.29	**GGGTAG**	**WSSWSW**	−15.10 (4.66)	−25.11	**AGCAGT**
**SSWSSS**	−37.08 (2.39)	−41.00	**GCACGC**	**WWWSSS**	−15.01 (3.75)	−25.22	**ATACGG**
**WWSWSW**	−37.06 (3.34)	−44.27	**AAGAGT**	**SWSSWS**	−15.01 (4.27)	−22.33	**CACCAG**
**WWSWWS**	−37.04 (2.23)	−42.15	**AAGTTG**	**WSSSWS**	−15.00 (3.64)	−25.23	**AGGGTC**
**WWSSSS**	−37.04 (2.23)	−42.27	**AAGGGG**	**SSWWWS**	−14.89 (3.88)	−22.08	**GCTTAG**
**WSWSSS**	−37.03 (2.66)	−43.21	**AGACCG**	**WSWWSS**	−14.87 (2.70)	−22.29	**TGTTCC**
**WSSWWW**	−36.96 (2.29)	−41.35	**AGGAAA**	**SSWWSS**	−14.86 (2.84)	−20.45	**CGAAGG**
**SSWSWS**	−36.85 (2.68)	−42.59	**GGACTG**	**WSWWWW**	−14.74 (3.13)	−18.83	**AGTATA**
**WSSSSW**	−36.81 (2.38)	−40.87	**ACGCGT**	**WWSSSS**	−14.63 (3.94)	−21.87	**ATCGGC**
**WSWSSW**	−36.71 (2.10)	−41.49	**TGAGGA**	**SWWSSS**	−14.62 (3.14)	−21.51	**CATCCG**
**WSSSWS**	−36.62 (2.19)	−40.55	**AGCGTC**	**WWSSSW**	−14.62 (3.20)	−22.91	**AAGGGT**
**WSWWSW**	−36.59 (2.56)	−41.18	**AGAAGT**	**WSWWWS**	−14.56 (2.98)	−22.72	**TCATTG**
**WWSWWW**	−36.47 (1.91)	−39.94	**ATGTTA**	**SSSWWS**	−14.51 (2.66)	−19.82	**GGGTAG**
**SWWSWS**	−36.44 (3.33)	−41.99	**GAAGTC**	**SWWWWS**	−14.48 (3.29)	−22.39	**GATATG**
**WSWSWS**	−36.34 (3.19)	−41.94	**ACAGTG**	**SWSWWS**	−14.38 (2.97)	−18.72	**GACAAG**
**WWSSSW**	−36.05 (2.83)	−41.93	**TAGCCA**	**WWSSWS**	−14.36 (3.83)	−26.00	**TACGAC**
**WWWSWS**	−35.86 (3.38)	−41.59	**TAACTC**	**WWWWSS**	−14.35 (3.07)	−22.83	**TTAACG**
**SSWWSS**	−35.79 (2.06)	−41.18	**GCAACG**	**SWSSSS**	−14.28 (4.52)	−26.11	**GAGGGG**
**WWSSWW**	−35.74 (2.17)	−39.62	**AACGTA**	**WSSWWS**	−14.02 (3.10)	−20.15	**ACCTAC**
**WSWWSS**	−35.59 (2.58)	−41.05	**AGTTCG**	**WSSSWW**	−14.01 (2.56)	−17.09	**ACGGTA**
**WSWWWW**	−35.16 (3.00)	−40.00	**AGTTTA**	**SSSSWS**	−13.98 (4.40)	−22.68	**GGGCTG**
**WWWSSW**	−35.14 (2.55)	−42.01	**AAACCT**	**WSWWSW**	−13.97 (2.36)	−19.40	**TGTTGA**
**SWWSSS**	−35.13 (2.52)	−39.57	**GAACCC**	**WWSWWS**	−13.93 (3.51)	−23.04	**AAGAAG**
**WWWSWW**	−35.06 (2.63)	−39.44	**AATGTT**	**WSSSSW**	−13.91 (4.71)	−23.66	**ACGGGT**
**WWWSSS**	−34.93 (2.14)	−39.43	**AAACGG**	**WWWWWS**	−13.86 (3.07)	−21.03	**TATAAG**
**WSSSWW**	−34.77 (2.58)	−39.61	**AGCCTA**	**WSSSSS**	−13.71 (4.14)	−26.54	**AGGGGC**
**SSWWWS**	−34.45 (2.59)	−39.13	**GCTTAG**	**WWWWSW**	−13.61 (3.46)	−20.26	**TATTGA**
**WWWWSW**	−34.32 (2.84)	−40.54	**AAAAGT**	**WWWSSW**	−13.47 (3.69)	−21.40	**TTTCCA**
**WSWWWS**	−34.25 (2.64)	−40.14	**AGATAG**	**WWSWWW**	−13.38 (3.44)	−19.74	**ATGTTA**
**SWWWSS**	−33.78 (2.37)	−38.47	**GAAACC**	**WSSWWW**	−13.27 (4.01)	−17.57	**AGCATA**
**WWWWSS**	−33.26 (2.26)	−38.85	**TAAACC**	**WWSSWW**	−13.06 (3.12)	−21.35	**ATGGTA**
**SWWWWS**	−31.90 (2.05)	−36.60	**CAAAAG**	**SSSSSS**	−12.78 (3.86)	−20.21	**GCGCCG**
**WWWWWS**	−31.82 (2.63)	−39.52	**TTAAAG**	**WWWWWW**	−12.50 (3.93)	−21.35	**TATATA**
**WWWWWW**	−31.76 (2.17)	−38.93	**AAAAAA**	**WWWSWW**	−11.90 (3.24)	−22.18	**ATAGTA**

* Unlike DNA sequence, mask sequences represent the sequence of base pairs AT&TA and GC&CG denoted as W (weak) and S (strong), respectively, according to standard IUPAC nomenclature [41].

Analysis of the promoter nucleotide sequences of genes involved in the pathogenesis of AD (*CASP3*, *TP53*, *SOD2*, *GPX1*, *PPARA*, *PPARG*, *NES*, *GAP43*, *SUMO1*, *APOE* and *IGF1*) showed the presence of DNA sequences found in low-energy complexes using molecular docking of the EDR peptide. For instance, the mask sequence SWSWSS with top ICM score (−38.6) for binding of the EDR peptide was found multiple times in the promoter regions of the *CASP3*, *NES GAP43* (3 times) and *APOE* (2 times) genes. Occurrence of other low-energy DNA mask sequences in promotors of the genes involved in AD is listed in Table 2.

The similar values of the top ICM scores for DNA sequences of six nucleotides and the EDR peptide indicate the ability of this tripeptide to regulate the expression of genes involved in the pathogenesis of Alzheimer’s disease.

## 3. Discussion

Neuroprotective peptide 5-oxo-PRP appeared in the rat brain tissues after intravenous and intranasal administration after 30 and 10 min, respectively [42]. Neuroprotective tripeptide GPE, which is the N-terminal fragment of IGF-1, crosses the blood–brain barrier when administered intravenously to rats. The accumulation of this peptide in all brain parts was registered within 4 h after administration [43]. RER (NH2-D-Arg-L-Glu-L-Arg-COOH) tripeptide, which is believed to have a neuroprotective effect in the early stages of AD, also crosses the blood–brain barrier [44]. Apart from that, it has been suggested that the synthetic snake-venom-based peptide p-BTX-I (Glu-Val-Trp), which potentially has a neuroprotective effect in Parkinson’s disease, can also cross the blood–brain barrier in terms of its physicochemical characteristics [39]. Analysis of the literature data suggests that neuroprotective tripeptides successfully cross the blood–brain barrier in many cases. It is possible that the studied EDR and KED tripeptides can also cross the blood–brain barrier. This hypothesis is supported by the neuroprotective effects of EDR and KED peptides in the in vivo AD model presented in this study.

The early manifestations of AD closely correlate with the synaptic loss [5]. At the same time, functional changes in synapses are reflected in dendritic spine morphology [9]. It was previously demonstrated that EDR and KED peptides prevent dendritic spine loss in an in vitro AD model [30]. To confirm this finding in vivo, we generated 5xFAD-M mice to analyze morphological changes in the CA1 region of the hippocampus. The EDR peptide increased the total CA1 dendritic spines density in the 5xFAD-M mice. This result may indicate the ability of the EDR peptide to stimulate forming new synaptic connections or to prevent the disruption of the newly formed synaptic sites. The KED peptide prevented the elimination of mushroom dendritic spines, which are particularly vulnerable in the case of AD.

In our study, neuroplasticity impairment was well correlated with the decrease in the mushroom spine number. This was consistent with the literature data on phenotypic differences in 5xFAD and wild-type mice [45]. Systematic administration of EDR and KED peptides in 5xFAD mice for 2 months revealed a tendency for the LTP increase in the CA1 region of the hippocampus. In addition, a pronounced trend of LTP increase in the 5xFAD mice with the KED peptide effect suggested that this peptide may be able to restore neuroplasticity at the early stages of AD. Thus, a systematic KED peptide administration provided a positive trend to the neuroplasticity restoration and prevented the elimination of functional synaptic connections, which in turn maintained the stability of mushroom spines. Our data are consistent with the EDR and KED peptides neuroprotective effects observed in vitro. However, in the AD in vitro model, the EDR peptide demonstrated a more pronounced neuroprotective action than the KED peptide [30]. The neuroprotective effect of the KED peptide may be related to its ability to prevent developing neurovascular degeneration, which has been demonstrated in the 5xFAD mice AD model [46]. In addition, it was established that the KED peptide has contributed to the endothelial cell growth factor (VEGF) recovery and endothelin-1 expression in the aortic endothelial cell culture received from patients with atherosclerosis [47,48]. At the same time, neuron-specific VEGF hyperexpression in the APP/Ps1 mouse AD model partially corrected the cerebral vessel loss and restored cognitive impairment [49]. In a comparative study on the efficacy of the KED and EDR peptides’ oral administration in patients of different ages with chronic polymorbid and organic brain syndromes, a greater efficacy also has been observed in the KED peptide. This was manifested in a faster recovery of cognitive functions and increased patients’ biological age [50]. This hypothesis explains the differences in the expressiveness of EDR and KED peptides effects in in vitro [30] and in vivo AD models.

We have demonstrated sex-based differences in neuroprotective effects of peptides in the 5xFAD-M mice model of AD. For example, EDR and KED peptides have contributed to the restoration of dendritic spines in 5xFAD males, which may be related to initially more pronounced AD pathology in male mice. Insofar as EDR and KED peptides are bioregulators, their effects can be modulated depending on the initial level of pathology, as previously shown for the KED and other short peptides [51,52].

It was previously shown that the KED peptide regulated the expression of cell aging and apoptosis of genes [53] and proteins (nestin, GAP-43) of neuronal differentiation [35]. In this study, additional molecular epigenetic points of peptides neuroprotective properties were discovered. Low-energy complexes of EDR and KED peptides with a 6-nucleotide dsDNA sequence were calculated using the molecular modeling method. These complexes are often found in the promoters of the CASP3, NES, GAP43, APOE, SOD2, PPARA, PPARG, GDX1 genes. The regulation of the expression of these genes by tripeptides may reflect the molecular mechanism of their neuroprotective activity. Interestingly, protein products of these genes are involved in neuroplasticity, synaptic pathology and cognitive impairments in AD.

A growth-associated phosphoprotein (GAP-43) gene expression in mature neurons is functionally important for the structural remodeling of synapses as required for learning and establishing new memory [54]. GAP-43 remains highly expressed in areas of the adult brain implicated in learning and memory, including the neocortex and hippocampus [55]. Hippocampal GAP-43 immunoreactivity was observed in dystrophic neurites correlated with aberrant sprouting [56,57], which is characteristic of synaptic pathology in AD [55]. In support of GAP-43 importance in information storage, heterozygous GAP-43 knockout mice (GAP-43^+/−^) exhibit a selective impairment in contextual memory [58].

Apolipoprotein E (ApoE) participates in the transport of plasma lipids and in the redistribution of lipids among cells. ApoE is implicated in the regeneration of synaptic circuitry after neural injury [59]. It was established that different apoE isoforms might produce differences in synaptic structure and function that make neuronal cells more susceptible to the toxic insults that occur with AD [60]. Mice expressing the human APOE ε4 allele have fewer dendritic spines than mice expressing the ε2 or ε3 alleles [59,61,62]. One study in humans observed a gene dose-dependent effect of APOE ε4 on dendritic spine density among Alzheimer’s disease patients and found that among cognitively normal individuals, ε4 carriers had fewer spines than noncarriers [59].

The intermediate filament protein nestin is a marker of neural stem cells in neurogenesis. It was found that the neurogenesis in the hippocampal dentate gyrus of adult nestin-deficient (Nes−/−) mice was increased. In behavioral studies, nestin deficiency was associated with impaired long-term memory [63]. A large number of studies have shown that the decrease of hippocampal neurogenesis may be closely related to the cognitive dysfunction caused by AD, and the increase of neurogenesis can improve the spatial memory of experimental animals [64].

Caspase-3 activity is essential for regulating spine density and dendrite morphology. Caspase-3 knockout mice have increased spine density and altered miniature excitatory postsynaptic currents, confirming a physiological involvement of caspase-3 in regulating spines in vivo. It was shown that caspase-3 is necessary for NMDA-receptor-dependent long-term depression that is associated with spine elimination [65]. In spines, caspase-3 activated calcineurin, which in turn triggered dephosphorylation and removal of the GluR1 subunit of AMPA-type receptor from postsynaptic sites. These molecular modifications led to alterations of glutamatergic synaptic transmission and plasticity and correlated with spine degeneration and a deficit in hippocampal-dependent memory. caspase-3 triggers early synaptic dysfunction in a mouse model of Alzheimer’s disease [66]. Compared to controls, AD patients exhibited significant increases in synaptic procaspase-3 and active caspase-3 expression levels [67].

Peroxisome-proliferator-activated receptor alpha (PPARA) downregulation may decrease antioxidative and anti-inflammatory processes and could be responsible for the alteration of fatty acid transport, lipid metabolism and disturbances of mitochondria function in the brain of AD patients. Specific activators of PPAR-α may be important for improved brain cell metabolism and cognitive function in neurodegenerative disorders, including AD [68,69]. Several studies have reported that PPARG agonists prevent the impairment of synaptic plasticity by increasing BDNF expression and dendrite spine density in hippocampal neurons treated with A*β* oligomers and A*β* injected rats [70,71]. At the same time, PPARG activation can simultaneously promote mitochondrial functions, improve metabolic and energy regulation, modulate neuroinflammation, stimulate axonal growth and myelination, and clear toxic A*β* from the brain [72,73].

Increasing evidence suggests that oxidative stress that is normally associated with aging is a prominent and early feature of AD and plays a role in its pathogenesis [74]. At the same time, mitochondrial dysfunction and/or endogenous oxidative stress is a prerequisite for neuronal loss in AD [75]. Thus, levels of oxidative markers, including glutathione peroxidase (GPx) and superoxide dismutase (SOD), in mitochondrial and synaptosomal fractions of postmortem frontal cortex declined significantly and correlated with Mini-Mental Status Examination scores in subjects with mild cognitive impairment, mild/moderate AD and late-stage AD. Oxidative stress was more localized to the synapses, with levels increasing in a disease-dependent fashion. These correlations demonstrate the involvement of oxidative stress in AD-related synaptic loss [74].

Thus, the neuroprotective effect of EDR and KED peptides in AD can be defined by their ability to prevent dendritic spine elimination and neuroplasticity impairments at the epigenetic level. We suppose that the EDR peptide effect is associated with the impact on the neuronal component of the brain. At the same time, the KED peptide affects the two levels of AD pathogenesis: neuronal loss and cerebral vessels endothelial dysfunction (Figure 12). The obtained results suggest that further research of EDR and KED peptides may be promising for developing a neuroprotective agent aimed at prevention and treatment of the early stage of AD.

## 4. Materials and Methods

Experimental design can be seen in the Figure below (Figure 13).

### 4.1. Tripeptides

Tripeptides EDR and KED (“GARMONIA”, Russia) were diluted in a physiological solution with the content of an amino acid complex of 0.1 mg/mL. Tripeptides were administered intraperitoneally in a 400 μg/kg concentration once a day for 2 months before electrophysiological or morphological experiments.

This concentration and method of administration of the peptides were chosen based on the analysis of the literature [77] and preliminary experiments.

### 4.2. Animals

The breeding colony of 5xFAD (B6SJL-Tg (APPSwFlLon, PSEN1*M146L*L286V) 6799Vas/Mmjax, #006554) mice with wild-type mice (WT) of the same strain (B6SJL background) were established and used for the neuroplasticity study from 4 months of age. The experimental groups were as follows: 1—WT mice treated with physiological solution (*n* = 10; 5 males, 5 females); 2—5xFAD mice treated with physiological solution (*n* = 10; 5 males, 5 females); 3—5xFAD mice treated with EDR peptide (*n* = 10; 5 males, 5 females); 4—5xFAD mice treated with the KED peptide (*n* = 10; 5 males, 5 females).

For the peptide effects’ study on spine morphology, a 5xFAD-M-line has been developed by cross-breeding of 5xFAD and M mice (Tg(Thy1-EGFP)MJrs/J, C57BL/6J background, #007788), obtained from the Jackson Laboratory (USA). The experimental groups were as follows: 1—M mice treated with physiological solution (*n* = 10; 5 males, 5 females); 2—5xFAD-M mice treated with physiological solution (*n* = 10; 5 males, 5 females); 3—5xFAD-M mice treated with EDR peptide (*n* = 10; 5 males, 5 females); 4—5xFAD-M mice treated with the KED peptide (*n* = 10; 5 males, 5 females).

All animals were maintained in a vivarium, four-five per cage with a 12 h light/dark cycle, and provided with standard food and water ad libitum.

### 4.3. Genotyping

Genomic DNA was isolated by the tail tip. The priming pairs for determining APP and PSEN1 transgenes were APP-oIMR3610-F (AGG ACT GAC CAC TCG ACC AG), APP-oIMR3611-R (CGG GGG TCT AGT TCT GCA T) and PSEN1-oIMR1644-F (AATAGAGAACGGCAGGAGCA), PSEN1-oIMR1645-R (GCCATGAGGGCACTAATCAT), respectively. The priming pair for determining GFP transgene was M-line-15731-R (CGG TGG TGC AGA TGA ACT T), M-line-16072-F (ACA GAC ACA CAC CCA GGA CA). The internal control primers were oIMR7338-F (CTA GGC CAC AGA ATT GAA AGA TCT), oIMR7339-R (GTA GGT GGA AAT TCT AGC ATC C).

### 4.4. Acute Hippocampal Slices Preparation

Four-month-old 5xFAD and WT mice were anesthetized by intraperitoneal (Zoletil^®^ Virbac+xylazine) injection. After verifying the sufficient depth of anesthesia, mice were perfused with carbogenated (95% O_2_/5% CO_2_) artificial cerebrospinal fluid (aCSF) with N-methyl-D-glucamine diatrizoate (NMDG) (solution 1: 92 mM NMDG, 2.5 mM KCl, 1.25 mM NaH_2_PO_4_, 30 mM NaHCO_3_, 20 mM 4-(2-hydroxyethyl)-1-piperazineethanesulfonic acid (HEPES), 25 mM D-glucose, 2 mM thiourea, 5 mM sodium ascorbate, 3 mM sodium pyruvate, 0.5 mM CaCl_2_, 10 mM MgSO_4_) [41] and decapitated. The mice brain was rapidly removed with subsequent cerebellum and cerebral hemisphere dorsal surface removal. The brain was fixed with superglue on the flat surface in a horizontal plane. Horizontal 400 um brain slices were prepared in ice-cold solution 1 using a vibratome VT1000S (Leica, France). Hippocampus was isolated from each slice. The slices were incubated on a nylon mesh attached to a glass beaker holding aCSF (Solution 2:92 mM NaCl, 1.25 mM NaH_2_PO_4_, 30 mM NaHCO_3_, 20 mM HEPES, 25 mM D-glucose, 2 mM thiourea, 5 mM sodium ascorbate, 3 mM sodium pyruvate, 2 mM CaCl_2_, 2 mM MgSO_4_) [41] in a temperature-controlled water bath (35 °C) for 1 h. Next, hippocampal slices were transferred to the recording chamber with a constant flow (5 mL/min) of carbogenated recording aCSF (solution 3:119 mM NaCl, 2.5 mM KCl, 1.25 mM NaH_2_PO_4_, 24 mM NaHCO_3_, 5 mM HEPES, 12.5 mM D-glucose, 2 mM CaCl_2_, 2 mM MgSO_4_) at room temperature. Hippocampal slices were kept for 15–20 min before the electrophysiological study [78]. One to four slices from each mouse were used in the experiment.

### 4.5. Electrophysiology

fEPSPs were recorded from stratum radiatum of CA1 using single glass microelectrodes (0.2–0.3 MΩ) filled with Solution 3. fEPSPs were evoked by stimulation of the Schaffer collaterals using bipolar electrodes (double-twisted 0.05 mm diameter chrome thread) placed in the stratum radiatum at the CA1–CA2 border. The stimulation was performed with rectangular pulses (duration, 0.1 ms) every 20 s by increasing the current intensity from 10 to 300 μA. The stimulus intensity used in the experiment was chosen such that the amplitude of the fEPSPs would be 40–50% of the amplitude, at which the population spike was first detected [78,79]. For each fEPSP, the amplitude and slope of the rising phase at a level of 20–80% of the peak amplitude were measured. The LTP was induced by high-frequency stimulation (HFS) protocol if stable baseline fEPSP amplitude had been recorded for 20 min. The fEPSPs were recorded after LTP induction for 60 min. fEPSPs were registered using an amplifier MultiClamp 700B (Molecular Devices, San Jose, CA, USA) with MultiClamp™ Commander software, a digitizer Digidata 1440A series (Molecular Devices, USA) and Clampex program of pClamp 10.7 software (Molecular Devices, USA). Data were analyzed by Clampfit 10.7 program (Axon Instruments). The baseline fEPSPs and the potentiated fEPSPs (recorded 50–60 min after HFS) were averaged separately to measure LTP. The plasticity value was calculated as the ratio of the slope of the rising phase in the averaged potentiated and baseline fEPSPs [78].

### 4.6. Fixed Brain Slices Preparation

5 month-old 5xFAD-M-line and M-line mice were anesthetized by intraperitoneal Urethane (250 mg/mL in 0.9% NaCl, Sigma-Aldrich, St. Louis, MO, USA) injection. After verifying the sufficient depth of anesthesia, mice were perfused (3 mL/min) with 10–20 mL PBS and 30–50 mL 4% paraformaldehyde (PFA) and decapitated. The mice brain was rapidly removed and placed in a 4% PFA for post-fixation for 1 week at 4 °C. Fixed 40 um brain slices were prepared using Lancer Vibratome Series 1000 Sectioning System 054,018 (USA) in 1x PBS. Prepared slices were placed under Aqua Poly/Mount (Polysciences, Inc. cat# 18606) mounting medium for a subsequent dendritic spine morphology analysis.

### 4.7. Dendritic Spine Morphology Analysis

Dendritic spine morphology was analyzed from Z-stack optical section with laser scanning confocal microscope ThorLabs with 100× objective lens (UPlanSApo; Olympus, Tokyo, Japan). The maximal resolution of each image in Z-stack was 1024 × 1024 pixels, with 0.067 μm/pixel, and averaged six times. The total Z volume was 4–8 μm imaged with a Z interval of 0.2 μm. The microphotographs obtained were processed in DeconvolutionLab plugin by Richardson–Lucy algorithm with a 0.02 regularization term and the point spread function established in the Huygens program. 10–15 fragments of CA1 second apical dendrites of each mouse were used for quantitative analysis. Quantitative analysis for dendritic spines was performed using the freely available NeuronStudio software package [80] as described [81]. To classify the shape of the neuronal spines, we adopted an algorithm from a published method [45]. In the classification of spine shapes, we used the following cutoff values: aspect ratio for thin spines (thin ratio) = 2.5, head-to-neck ratio (HNRcrit) = 1.3, and head diameter (HDcrit) = 0.35 μm. For neuron morphology characterization, we used spine density (SD) calculated as ratio of total spines number to 10 um of dendritic length and mushroom (MS), thin (TS) and stubby (SS) spines number calculated relatively to the total spines number and expressed in%.

### 4.8. Statistical Analysis

Statistical analysis was performed by Clampfit (Axon™pCLAMP™ 10 Electrophysiology Data Acquisition and Analysis Software) and Statistica 12 programs. Statistical significance was assessed by Kruskal–Wallis test, Jonckheere’s trend test in neuroplasticity experiment and Student’s *t*-test, one-way ANOVA following Dunnett’s post hoc test in dendritic spine analysis. The results were presented as mean ± SEM. Significance level was 0.05, 0.01, 0.001.

### 4.9. Molecular Modeling

#### 4.9.1. Structure Preparation

Spatial structures of EDR and KED tripeptides with free N- and C-termini were generated with the ICM-Pro software package (Molsoft LLC, La Jolla, CA, USA). Structures of dsDNA consisting of all possible combinations of 6 base pairs having unique spatial structures (totally 2080 sequences) were generated in the central part of 14-mers flanked by four AT nucleotide pairs at both 3′ and 5′ termini. The resulting dsDNA structures were energy minimized in the ICMFF force field of the ICM-Pro software package using default settings of energy parameters for van der Waals, electrostatic, hydrogen bonding, torsion energy interactions and solvation free energy [82].

#### 4.9.2. Virtual Ligand Screening and Analysis of the Results Obtained

The virtual screening of the ligands in the target dsDNA binding pocket was performed using the ICM-Dock method, the ICMFF force field and ICM standard protocols for docking of flexible ligands as implemented in the DockScan utility of ICM-Pro software package (Molsoft LLC) [83]. The calculations were done using supercomputer facilities of Petersburg Institute of Nuclear Physics NRC “Kurchatov Institute”. The search for peptide conformations in DNA–ligand complexes was carried out with the highest thoroughness corresponding to the number of free torsion angles of ligand (thorough = 30), which was selected on preliminary tests of the reproducibility of the docking results as described earlier [84]. The most energetically favorable positions of all ligands in every receptor under consideration were selected for further analysis.

Nucleotide sequences of gene promoters involved in AD pathogenesis (CASP3, TP53, SOD2, GPX1, PPARA, PPARG, NES, GAP43, SUMO1, APOE, IGF1) [31,35,51] were obtained from the EPD (the Homo sapiens curated promoter database). For the initial sequences of the studied promoters, complementary and reverse sequences in the 5′ −> 3′ and 3′ −> 5′ directions were compiled. The search in the nucleotide sequences of gene promoters was carried out for the best DNA sequences found in peptide-DNA complexes using molecular docking.

## 5. Conclusions

The EDR and KED peptides prevent dendritic spine loss in in vitro and in vivo AD models. To access dendritic spine morphology in vivo, we utilized the advantage of M-line mice, which expressed GFP in hippocampal neurons and generated the 5xFAD-M mice. Peptide KED provided the trend to normalization of LTP in 5xFAD mice. The EDR and KED peptides produced a bioregulatory effect on the morphology of the CA1 neuron dendritic spine in AD. The exact mechanism of action of these peptides is unknown. The possible interaction between short peptides and dsDNA was predicted earlier [85]. Using methods of molecular modeling and docking of peptides in dsDNA with arbitrary sequences, we have shown that there are almost identical hexanucleotide sequences in the lowest-energy complexes of EDR peptide with dsDNA, which are often found in the promoter regions of such genes as *CASP3*, *NES*, *GAP43*, *APOE*, *SOD2*, *PPARA*, *PPARG*, *GDX1*. The neuroprotective effect of EDR and KED peptides in AD can be defined by their ability to prevent dendritic spine elimination and neuroplasticity impairments at the molecular epigenetic level. The EDR and KED peptide can be considered as novel promising agents for future treatment of early AD stages.

## Figures and Tables

**Figure 1 pharmaceuticals-14-00515-f001:**
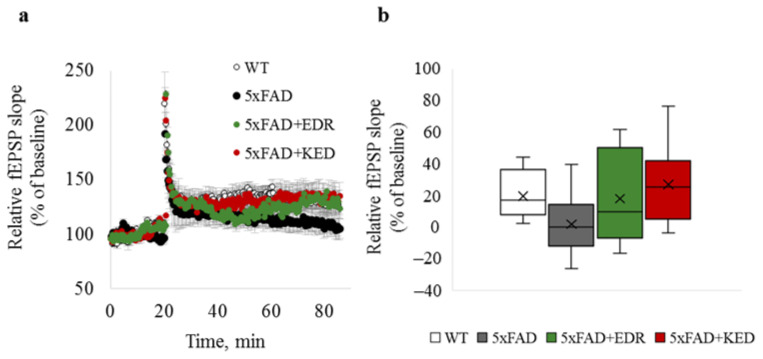
LTP in hippocampal brain slices of WT (white) and 5xFAD mice injected with a physiological solution (black), EDR peptide (green) and KED peptide (red). (**a**) Time course of fEPSP slope 20 min before and 60 min after LTP induction; (**b**) box plot illustrating plasticity value estimated by a relative fEPSP slope 50 min after LTP induction. The middle bar in the boxes represents the median; the cross in the boxes represents the mean; the lower and upper ends of the boxes represent the first and third quartiles, respectively. “Whiskers” represent values within 1.5 times the interquartile range from the upper and lower quartile.

**Figure 2 pharmaceuticals-14-00515-f002:**
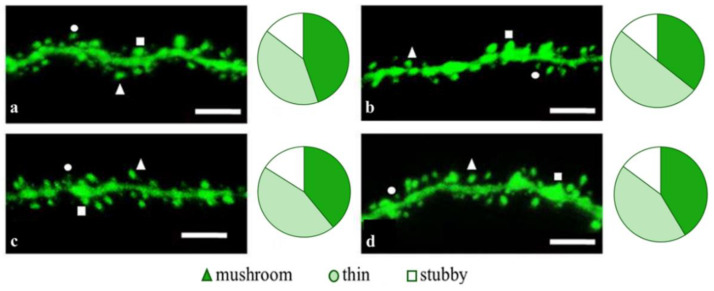
Confocal images of the CA1 secondary dendrites in 5 month-old M (**a**) and 5xFAD-M mice, injected with a physiological solution (**b**), EDR peptide (**c**), KED peptide (**d**). Confocal microscopy, ×100. Scale bar, 10 um.

**Figure 3 pharmaceuticals-14-00515-f003:**
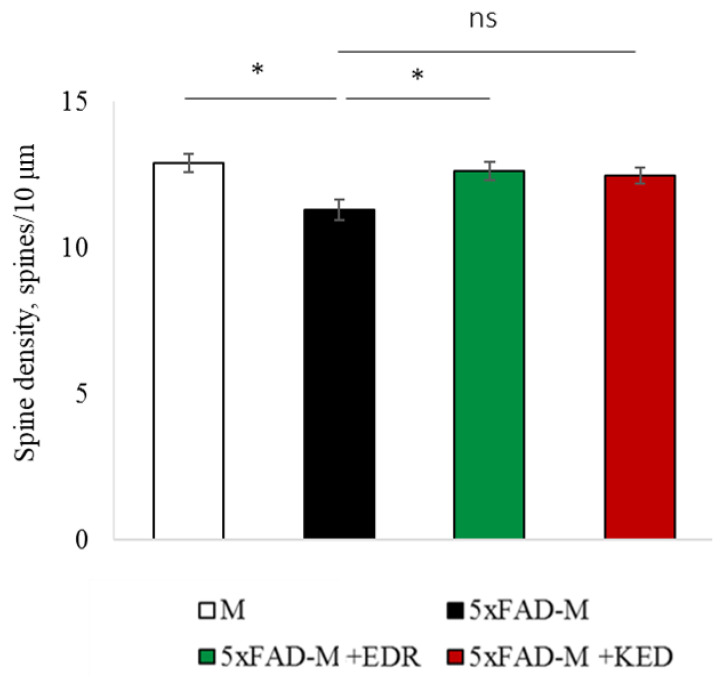
Spine density of CA1 secondary dendrites in 5 month-old M and 5xFAD-M mice, injected with physiological solution, EDR peptide, KED peptide. *—*p* < 0.05 compared to the 5xFAD-M mice, injected with physiological solution, ns—non-significant.

**Figure 4 pharmaceuticals-14-00515-f004:**
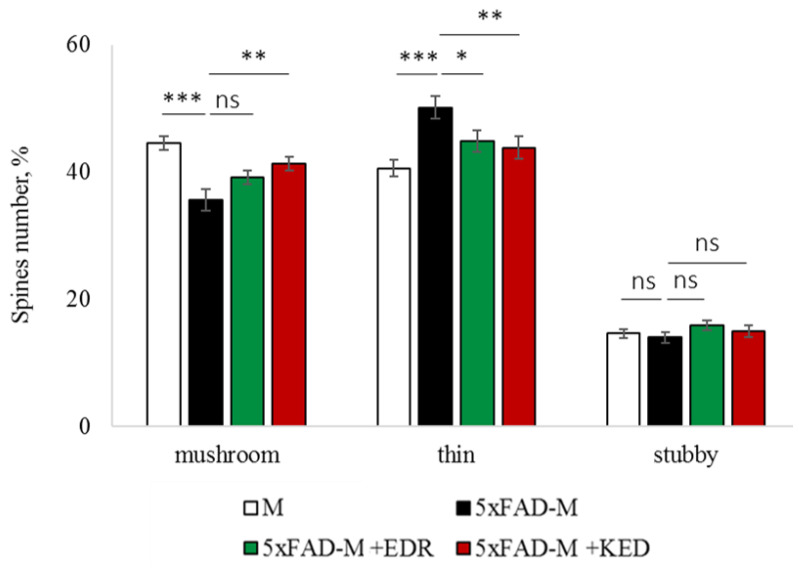
Relative spines number of CA1 secondary dendrites in 5 month-old M and 5xFAD-M mice, injected with physiological solution, EDR peptide, KED peptide. *, **, ***—*p* < 0.05, *p* < 0.01, *p* < 0.001 compared to the 5xFAD-M mice, injected with physiological solution, ns—non-significant.

**Figure 5 pharmaceuticals-14-00515-f005:**
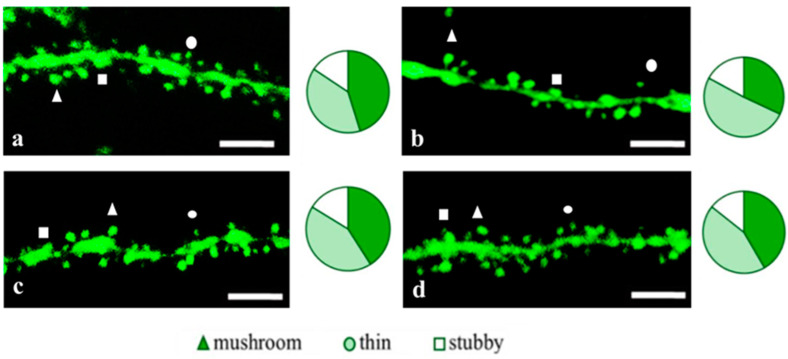
Confocal images of the CA1 secondary dendrites in 5 month-old male M-line (**a**) and 5xFAD-M mice, injected with a physiological solution (**b**), EDR peptide (**c**), KED peptide (**d**). Confocal microscopy, ×100. Scale bar, 10 μm.

**Figure 6 pharmaceuticals-14-00515-f006:**
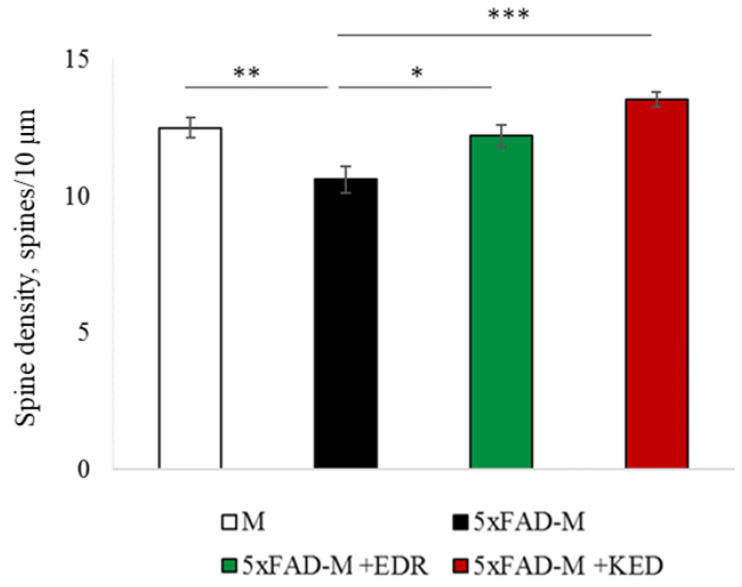
Spine density of CA1 secondary dendrites in 5 month-old male M-line and 5xFAD-M mice, injected with physiological solution, EDR peptide, KED peptide. *, **, ***—*p* < 0.05, *p* < 0.01, *p* < 0.001 compared to the 5xFAD-M mice injected with physiological solution, ns—non-significant.

**Figure 7 pharmaceuticals-14-00515-f007:**
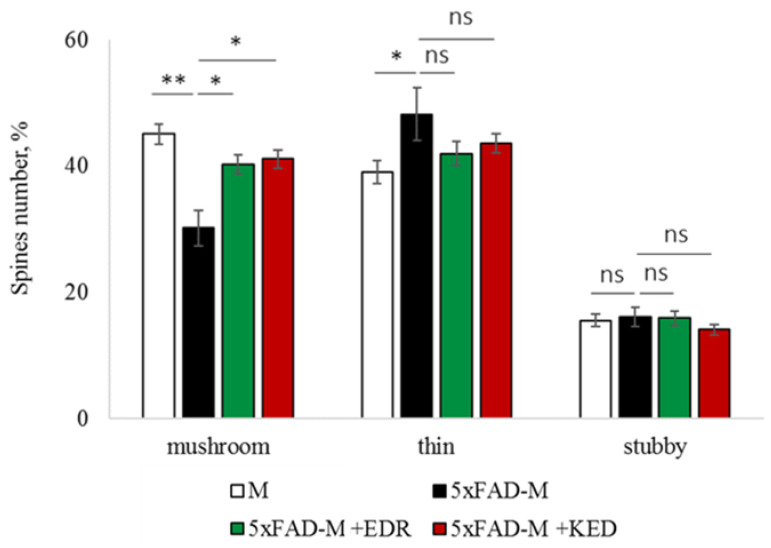
Relative spines number of CA1 secondary dendrites in 5 month-old male M-line and 5xFAD-M mice, injected with physiological solution, EDR peptide, KED peptide. *, **—*p* < 0.05, *p* < 0.01 compared to the control 5xFAD-M mice, injected with physiological solution, ns—non-significant.

**Figure 8 pharmaceuticals-14-00515-f008:**
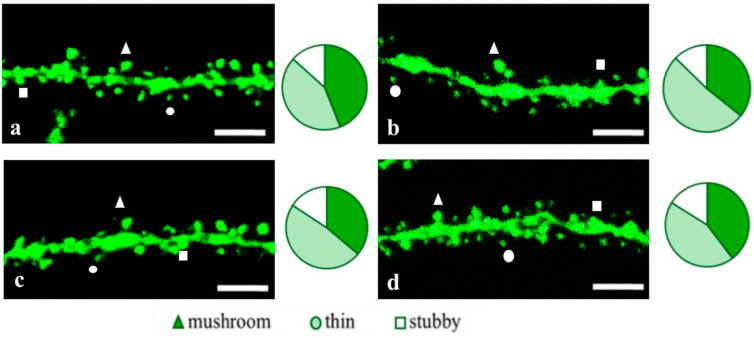
Confocal images of the CA1 secondary dendrites in 5 month-old female M (**a**) end 5xFAD-M mice injected with a physiological solution (**b**), EDR peptide (**c**), KED peptide (**d**). Confocal microscopy, ×100. Scale bar, 10 μm.

**Figure 9 pharmaceuticals-14-00515-f009:**
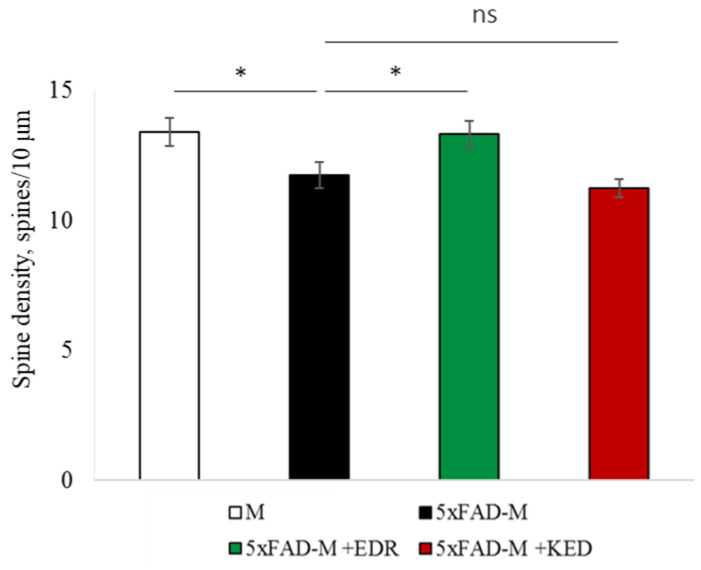
Spine density of CA1 secondary dendrites in 5 month-old female M and 5xFAD-M mice, injected with physiological solution, EDR peptide, KED peptide. *—*p* < 0.05 compared to the 5xFAD-M mice, injected with physiological solution, ns—non-significant.

**Figure 10 pharmaceuticals-14-00515-f010:**
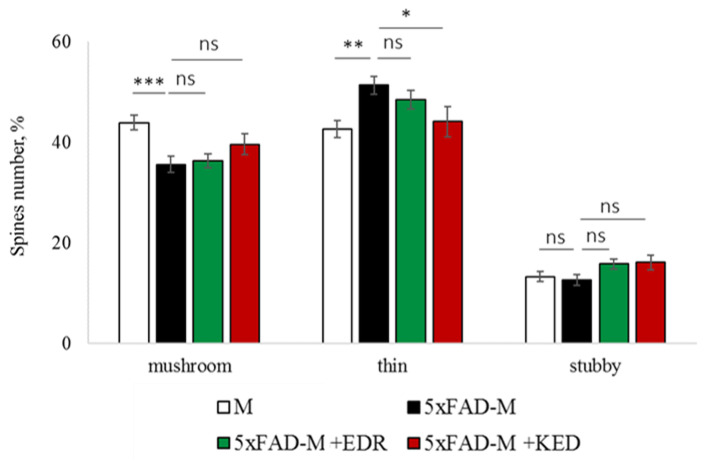
Relative mushroom spines number of CA1 secondary dendrites in 5 month-old female M and 5xFAD-M mice, injected with physiological solution, EDR peptide, KED peptide. *, **, ***—*p* < 0.05, *p* < 0.01, *p* < 0.001 compared to the control 5xFAD-M mice, injected with physiological solution, ns—non-significant.

**Figure 11 pharmaceuticals-14-00515-f011:**
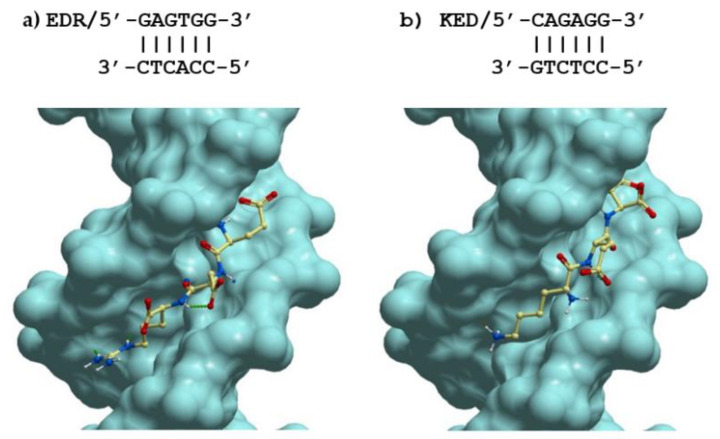
Lowest energy structures of complexes of the tripeptides (**a**) EDR and (**b**) KED with their best-fit dsDNA receptors in classic B-form.

**Figure 12 pharmaceuticals-14-00515-f012:**
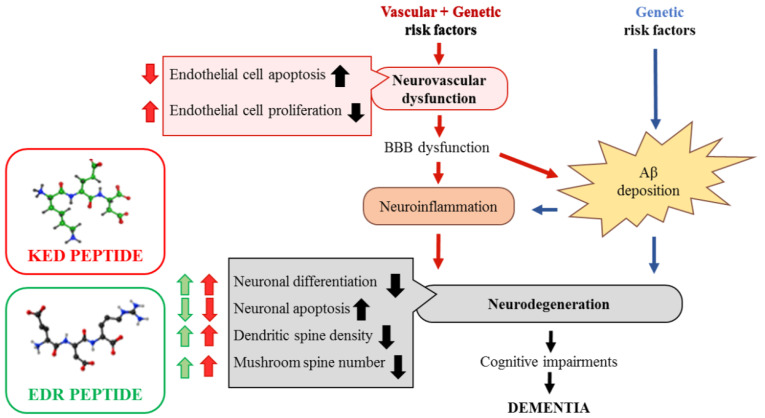
Proposed neuro-vasoprotective effects of the KED and EDR peptides in Alzheimer’s disease. Vascular factors (hypertension and diabetes) and/or genetic risk factors for AD, such as the ε4 allele of apolipoprotein E, can all lead to a neurovascular dysfunction in cerebral vessels. It leads to a blood–brain barrier (BBB) dysfunction and accumulation of blood-derived neurotoxic molecules in the brain, causing neuroinflammation. Additionally, within the amyloidogenic Aβ pathway, BBB dysfunction can disrupt Aβ clearance across the BBB and promote Aβ accumulation in the brain. These two ways can independently and/or synergistically lead to synaptic and neuronal dysfunction, neurodegeneration, cognitive impairments, which ultimately lead to dementia [76]. The EDR peptide improves neuronal differentiation, dendritic spine density and increases the number of mushroom spines, which are decreased in AD. The KED peptide provides decreased endothelial and neuronal cell apoptosis and increased endothelial cell proliferation, neuronal differentiation, dendritic spine density, and the number of mushroom spines in AD.

**Figure 13 pharmaceuticals-14-00515-f013:**
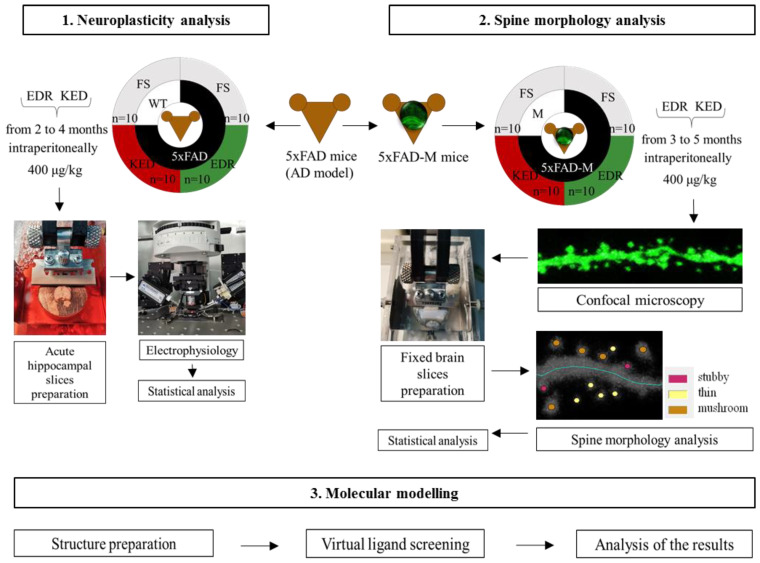
Experimental schedule. Note: FS—physiological solution.

**Table 2 pharmaceuticals-14-00515-t002:** Occurrence of low-energy DNA mask sequences in promotors of the genes involved in AD.

N	Mean ICM Score (STD)	DNA Mask Sequence	Gene	Frequency of DNA Mask Sequence Occurrence in Gene Promotor
1	−38.66	**WSSWSS**	*GPX1*	1 time
2	−38.20	**SWSSWS**	*APOE*	2 times
3	−38.02	**SSSWSS**	*PPARA*	3 times
			*GAP43*	4 times
4	−37.96	**WSSWSW**	*SOD2*	1 time
			*APOE*	1 time
5	37.89	**SWSWWS**	*SOD2*	1 time
			*PPARA*	2 times
			*GAP43*	1 time
6	−37.73	**SWSSSS**	*PPARA*	2 times
			*GAP43*	1 time
			*APOE*	2 times
7	−37.65	**WWSWSS**	*SOD2*	1 time
8	−37.45	**WSSWWS**	*PPARG*	1 time
9	−37.36	**WWSSWS**	*SOD2*	1 time
			*GPX1*	1 time

## Data Availability

All the data analyzed in this study are included in the published article. The primary data used for the analysis are available upon request from the corresponding author.

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
