# Peer review of "Neuroprotective Effects of Tripeptides—Epigenetic Regulators in Mouse Model of Alzheimer’s Disease"

_pharmaceuticals, 2021, doi:10.3390/ph14060515_

Round 1
Reviewer 1 Report
In this study, the authors showed that tripeptides KED and EDR prevented dendritic spine loss in 5xFAD Alzheimer model mice. In addition, they showed that the peptides have binding sites in promoter region of CASP3, NES, GAP43, APOE, SOD2, PPARA, PPARG, GDX1 genes by molecular modeling and docking. Although the results are generally interesting, I have several concerns.
- In the Introduction (line 46-48), the authors mentioned that the processing of APP by alpha-secretase and gamma-secretase results in the formation of a short beta-amyloid (Abeta 1-40) fragment that is not neurotoxic. This statement is wrong. Abeta 1-40 is produced by beta-secretase and gamma-secretase, but not alpha-secretase and gamma-secretase. In addition, not only Abeta 1-42 but also Abeta 1-40 is neurotoxic.
- In the Results, the authors showed the sex differences in spine density and spine numbers in peptide-treated 5xFAD mice. What about the sex differences in LTP?
- In the Materials and Methods, the experimental design is a little confusing. I suggest the authors to make a figure showing the experimental schedule.
- The authors should describe the reason why they chose the dose of tripeptides (400 ug/kg) as well as the route of administration.
- Do the tripeptides cross the blood-brain barrier? It would be better to add a discussion about the pharmacokinetic properties of the tripeptides.
Author Response
Thank you for reviewing our article. Your recommendations will help us improve the manuscript. Please see our comments below.
- In the Introduction (line 46-48), the authors mentioned that the processing of APP by alpha-secretase and gamma-secretase results in the formation of a short beta-amyloid (Abeta 1-40) fragment that is not neurotoxic. This statement is wrong. Abeta 1-40 is produced by beta-secretase and gamma-secretase, but not alpha-secretase and gamma-secretase. In addition, not only Abeta 1-42 but also Abeta 1-40 is neurotoxic.
You are absolutely correct and we agree that Abeta 1-40 is produced by beta-secretase and gamma-secretase. Of course, there is no doubt that not only Abeta 1-42 but also Abeta 1-40 is neurotoxic. As per your comment, we have made corresponding changes to the manuscript (lines 49-52).
- In the Results, the authors showed the sex differences in spine density and spine numbers in peptide-treated 5xFAD mice. What about the sex differences in LTP?
No sex differences in LTP have been revealed.
- In the Materials and Methods, the experimental design is a little confusing. I suggest the authors to make a figure showing the experimental schedule.
The figure showing the experimental schedule (Figure 13) has been added (lines 526-529). We hope it will contribute to the overall clarity of our manuscript.
- The authors should describe the reason why they chose the dose of tripeptides (400 ug/kg) as well as the route of administration.
This concentration and method of administration of the peptides were chosen based on the analysis of the article [Eremin KO, Kudrin VS, Saransaari P, Oja SS, Grivennikov IA, Myasoedov NF, Rayevsky KS. Semax, an ACTH(4-10) analogue with nootropic properties, activates dopaminergic and serotoninergic brain systems in rodents. Neurochem Res. 2005 Dec;30(12):1493-500. doi: 10.1007/s11064-005-8826-8. PMID: 16362768] and preliminary experiments. In the article by Eremin et al. rats were administered with Semax intraperitoneally at concentrations 0.15 and 0.6 mg/kg. During our preliminary experiments, it was revealed that a tripeptide concentration of 400 μg/kg is the most appropriate for studying the neuroplasticity and morphology of dendritic spines in mice in vivo.
We have added an explanation for choosing the dose and the route of administration of the peptides (lines 535-539).
- Do the tripeptides cross the blood-brain barrier? It would be better to add a discussion about the pharmacokinetic properties of the tripeptides.
We know the following about the pharmacokinetic properties of tripeptides: neuroprotective peptide 5-oxo-PRP was shown to appear in the rat brain tissues after intravenous and intranasal administration after 30 and 10 minutes, respectively [The penetration of 5-oxo-Pro-Arg-Pro into the brain and the major metabolic pathways of this peptide in the rat brain and blood at the intranasal and intravenous administration. Shevchenko KV, Nagaev IY, Shevchenko VP, Andreeva LA, Shram SI, Myasoedov NF.Dokl Biochem Biophys. 2017 Mar;473(1):151-154. doi: 10.1134/S1607672917020168]. Neuroprotective tripeptide GPE, which is the N-terminal fragment of IGF-1, crosses the blood-brain barrier when administered intravenously to rats. The accumulation of this peptide in all parts of the brain was recorded within 4 h after administration. [Neuroprotective effects of the N-terminal tripeptide of insulin-like growth factor-1, glycine-proline-glutamate (GPE) following intravenous infusion in hypoxic-ischemic adult rats. Guan J, Thomas GB, Lin H, Mathai S, Bachelor DC, George S, Gluckman PD.Neuropharmacology. 2004 Nov;47(6):892-903. doi: 10.1016/j.neuropharm.2004.07.002]. RER (NH2-D-Arg-L-Glu-L-Arg-COOH) tripeptide, which is believed to have a neuroprotective effect in the early stages of AD, also crosses the blood-brain barrier [Protection against Abeta-induced memory loss by tripeptide D-Arg-L-Glu-L-Arg. Mileusnic R, Lancashire C, Clark J, Rose SP.Behav Pharmacol. 2007 May;18(3):231-8. doi: 10.1097/FBP.0b013e32814fcde9]. Apart from that, it has been suggested that the synthetic snake-venom-based peptide p-BTX-I (Glu-Val-Trp), which potentially has a neuroprotective effect in Parkinson's disease, can also cross the blood-brain barrier in terms of its physicochemical characteristics [A synthetic snake-venom-based tripeptide (Glu-Val-Trp) protects PC12 cells from MPP(+) toxicity by activating the NGF-signaling pathway. Bernardes CP, Santos NAG, Sisti FM, Ferreira RS, Santos-Filho NA, Cintra ACO, Cilli EM, Sampaio SV, Santos AC.Peptides. 2018 Jun;104:24-34. doi: 10.1016/j.peptides.2018.04.012]. Analysis of literature data allows concluding that in a number of cases neuroprotective tripeptides successfully cross the blood-brain barrier. It is possible that EDR and KED tripeptides can also cross the blood-brain barrier. This hypothesis is supported by the neuroprotective effects of EDR and KED peptides in the in vivo AD model presented in this study.
We agree that it would be better to add the Discussion section with the data on the pharmacokinetics of tripeptides. Therefore, we made changes to lines 391-422 of the manuscript.
Reviewer 2 Report
A few minor corrections:
- Explain the abbreviation LPT
- Figure 1 a is hard to read specially in black and white printout. Would it be possible to remove the dots from the graph lines?
- Page 3, line 140 – remove the insertion of unnecessary text
- Description of results in 2.2. should be followed by exact figure numbers.
Author Response
Thank you for reviewing our article. Your recommendations will help us improve the manuscript. Please see our comments below.
- Explain the abbreviation LPT
LTP means long-term potentiation. The explanation has been added (lines 69-70).
- Figure 1 a is hard to read specially in black and white printout. Would it be possible to remove the dots from the graph lines?
Unfortunately, we cannot remove the dots from the graph, since each dot reflects an experimentally obtained number. In this regard, the graph itself is a set of dots, which form a line on the graph. If you don't mind, we would like to leave the graph as it is.
- Page 3, line 140 – remove the insertion of unnecessary text
Thank you for your comment. The unnecessary text has been removed.
- Description of results in 2.2. should be followed by exact figure numbers.
Section 2.2.contains links to pictures in lines 149, 156, 167, 174, 178.
Round 2
Reviewer 1 Report
The authors revised the manuscript properly.